# *APOA1/C3/A4/A5* Gene Cluster at 11q23.3 and Lipid Metabolism Disorders: From Epigenetic Mechanisms to Clinical Practices

**DOI:** 10.3390/biomedicines12061224

**Published:** 2024-05-31

**Authors:** Qianqian Xiao, Jing Wang, Luyun Wang, Hu Ding

**Affiliations:** 1Division of Cardiology, Departments of Internal Medicine, Tongji Hospital, Tongji Medical College, Huazhong University of Science and Technology, Wuhan 430030, China; xiaoqq9711@163.com (Q.X.); wangjing81531@163.com (J.W.); wangluyun@tjh.tjmu.edu.cn (L.W.); 2Hubei Key Laboratory of Genetics and Molecular Mechanisms of Cardiological Disorders, Wuhan 430030, China

**Keywords:** apolipoproteins, lipid metabolism disorders, epigenetics, molecular therapies, hyperlipidemia

## Abstract

The *APOA1/C3/A4/A5* cluster is an essential component in regulating lipoprotein metabolism and maintaining plasma lipid homeostasis. A genome-wide association analysis and Mendelian randomization have revealed potential associations between genetic variants within this cluster and lipid metabolism disorders, including hyperlipidemia and cardiovascular events. An enhanced understanding of the complexity of gene regulation has led to growing recognition regarding the role of epigenetic variation in modulating *APOA1/C3/A4/A5* gene expression. Intensive research into the epigenetic regulatory patterns of the *APOA1/C3/A4/A5* cluster will help increase our understanding of the pathogenesis of lipid metabolism disorders and facilitate the development of new therapeutic approaches. This review discusses the biology of how the *APOA1/C3/A4/A5* cluster affects circulating lipoproteins and the current progress in the epigenetic regulation of the *APOA1/C3/A4/A5* cluster.

## 1. Introduction

Dyslipidemia is a commonly encountered chronic condition in clinical practice involving abnormalities in plasma cholesterol, triglycerides (TGs), or both [1]. Dyslipidemia is also an important risk factor for atherosclerotic cardiovascular diseases (ASCVDs), such as coronary artery disease (CAD), stroke, and peripheral vascular disease [2]. Elevated levels of total cholesterol (TC), low-density lipoprotein cholesterol (LDL-C), TGs, and lipoprotein A, along with decreased high-density lipoprotein cholesterol (HDL-C), are major contributors to ASCVD risk [3]. Systemic metabolic diseases, including obesity and diabetes, are intimately associated with dyslipidemia [4,5]. Dyslipidemia leads to the intrahepatic accumulation of fat, resulting in non-alcoholic fatty liver disease (NAFLD), non-alcoholic steatohepatitis (NASH), and liver fibrosis [6,7]. Additionally, severe hypertriglyceridemia (HTG) results in critical clinical conditions, including acute pancreatitis and familial chylomicronemia syndrome (FCS) [8,9]. Therefore, in the primary and secondary prevention of lipid disorders, the treatment of dyslipidemia plays an important role [10].

The *APOA1/C3/A4/A5* gene cluster, located on chromosome 11q23.3, comprises a group of closely related apolipoprotein (APO) genes with interrelated functions, which are important modulators of lipoprotein transport and metabolism [11,12]. Epidemiological studies and fundamental experiments consistently demonstrate the central role of *APOA1/C3/A4/A5* in intestinal, plasma, and hepatic lipid homeostasis [12,13].

Recently, SNPs in *APOA1* have been widely utilized as predictive markers for CAD risk, and prospective studies of weight loss in obese patients have shown that the *APOA1* (rs670) gene displays significant effects on LDL cholesterol levels and insulin resistance [14,15,16]. Large-scale population genetic studies have shown that loss-of-function (LOF) mutations in the *APOC3* gene confer reduced very-low-density lipoprotein (VLDL) and non-HDL cholesterol levels, which are associated with a decreased risk of HTG and coronary atherosclerosis [17,18]. *APOA4* gene polymorphisms serve as important indicators for renal and cerebral vascular diseases [19,20]. Several meta-analyses have shown that SNPs in *APOA5* may be important contributors to the elevated risk of cardiovascular disease. Specifically, compared with carriers of the T allele, carriers of the *APOA5* rs662799-C allele have higher plasma TC and TG levels and lower plasma HDL-C levels [21,22]. Neighboring genes of the *APOA1/C3/A4/A5* gene cluster, including *BUD13*, *ZPR1*, and *SIK1*, have also been implicated in lipid metabolism [23,24]. SNPs in the *BUD13* and *ZPR1* genes have strong associations with elevated triglyceride levels in patients with metabolic syndrome [25]. *SIK1* is expressed in liver tissue and may affect lipid metabolism by regulating fatty acid synthesis and catabolism pathways [26]. In conclusion, the *APOA1/C3/A4/A5* cluster holds potential as a novel target for addressing lipid metabolism disorders (Figure 1).

The heredity of most organisms is based on the inheritance of genomic DNA [27]. Although the relationship between the genetic variability and lipid metabolism of *APOA1/C3/A4/A5* has been extensively elucidated, there remain functional changes and disease consequences that cannot be solely accounted for by gene mutations [11,12]. Notably, numerous studies have identified the importance of epigenetic processes in regulating *APOA1/C3/A4/A5* gene expression (Figure 2). Epigenetic modifications are heritable changes in gene expression caused by environmental factors without altering the DNA sequence. Epigenetic modifications mainly include DNA methylation, histone methylation, histone acetylation, RNA modification, and non-coding RNA [28]. This epigenetic control is critical to the physiological control of transcription and translation processes [27]. As epigenetic controls of *APOA1/C3/A4/A5* have been continually reported in recent years, numerous novel insights into diagnosis, treatment, and prognosis originate from epigenetics. Therefore, this paper comprehensively investigates the diverse mechanisms underlying epigenetic regulation in the *APOA1/C3/A4/A5* gene cluster and discusses emerging strategies for treating lipid metabolism disorders.

## 2. APOA1: Proteins with Therapeutic Potential for Lipid Metabolism Disorders

### 2.1. Function of APOA1

APOA1 is the major protein fraction of high-density lipoprotein (HDL) particles, playing a pivotal role in HDL synthesis [29] (Figure 1). HDL, commonly regarded as “good cholesterol”, exerts beneficial effects such as reducing cardiovascular risk by facilitating reverse cholesterol transport (RCT) [30,31]. HDL can help remove excess cholesterol from peripheral tissues to the liver, where it undergoes clearance and subsequent excretion via bile [32]. In addition, HDL acts as an inhibitor of atherosclerotic, inflammatory, and apoptotic properties, which, together, enhance its protective role against ASCVD [33,34].

*APOA1* is primarily produced by hepatocytes and then released into the bloodstream [29]. APOA1 interacts with the ATP-binding cassette transporter A1 (ABCA1) and triggers cholesterol efflux, promoting the transfer of intracellular cholesterol and phospholipid to nascent high-density lipoprotein (nHDL) [35,36,37]. Subsequently, APOA1 interacts with lecithin–cholesterol acyltransferase (LCAT) to activate LCAT and promote the conversion of nHDL to mature HDL [38]. Mature HDL can be selectively removed from plasma by the hepatic HDL receptor, scavenger receptor type BI (SR-BI). Cholesterol in HDL can also be transferred to TRLs, including very-low-density lipoprotein (VLDL) and LDL, under the action of cholesteryl ester transfer proteins (CETPs), and then it can be transported to the liver by low-density lipoprotein receptors (LDLRs) [36,37,39].

During the progression of atherosclerotic disease, vascular endothelial cells release pro-inflammatory signals, including intercellular adhesion molecule-1 (ICAM-1) and vascular cell adhesion molecule-1 (vCAM-1), which significantly contribute to the initiation of early lesions. These molecules are members of the immunoglobulin superfamily of cell adhesion molecules (CAMs) and mediate leukocyte adherence to endothelial cells during atherosclerosis and myocardial infarction [40]. Recent studies have demonstrated that the injection of APOA1 into rabbits can downregulate the expression levels of VCAM-1 and ICAM-1, thus exerting anti-inflammatory and anti-atherosclerotic effects [41]. APOA1 also has potent anti-inflammatory effects on macrophages through the rapid disruption of lipid rafts and the inhibition of the PI3K/AKT pathway, which significantly reduces macrophage chemotaxis [42]. Simultaneously, facilitated by the APOA1 binding protein (AIBP), APOA1 tightly binds to ABCA1 on the macrophage membrane, stabilizing the ABCA1 protein against COP9 signalosome subunit 2 (CSN2)-mediated degradation. This process prevents foam cell formation [43]. Together, these findings suggest that APOA1 has the ability to attenuate inflammatory cytokine production in endothelial cells and macrophages, thereby exhibiting anti-inflammatory properties.

### 2.2. Epigenetic Regulation and Therapeutic Potential of APOA1

The regulation of *APOA1* is complex and occurs at different stages of gene expression (Figure 2). *APOA1-AS*, the natural antisense strand of *APOA1*, induces the silencing of *APOA1* gene expression by recruiting lysine (K)-specific demethylase 1 (LSD1) to catalyze the demethylation of tri-methylated histone H3 lysine 4 (H3K4) in the promoter region of *APOA1*. However, the antisense oligonucleotide (ASO) targeting *APOA1-AS* leads to the upregulation of *APOA1* expression both in vitro and in vivo, suggesting that *APOA1-AS* is an efficient transcriptional regulator of *APOA1* [44]. Given its high specificity, stability, and efficiency, ASO targeting *APOA1-AS* may emerge as an effective lipid-lowering drug in the near future [45].

*APOA1-AS* is also crucial for regulating coronary atherosclerosis. In vascular smooth muscle cells (VSMCs) treated with oxidized low-density lipoprotein (ox-LDL), the expression of *APOA1-AS* is significantly upregulated. Conversely, the absence of *APOA1-AS* impairs the proliferation and migration of ox-LDL-VSMCs and stimulates cell apoptosis. Mechanistically, *APOA1-AS* recruits TATA-box binding protein-related factor 15 (TAF15) to stabilize SMAD family member 3 (SMAD3) mRNA, subsequently activating the TGF-β/SMAD3 signaling pathway to promote VSMC proliferation and migration. In addition, *APOA1-AS* inhibits VSMC apoptosis, highlighting its therapeutic potential for coronary atherosclerosis [46,47,48].

Cholesterol participates in reproductive processes by being a precursor for the synthesis of synthetic sex hormones. Disorders in lipid metabolism within sexual organs can result in fluctuations in sex hormone levels, subsequently impacting the growth or functionality of reproductive organs [49]. Bisphenol A, a widely utilized plasticizer found abundantly in aquatic environments, exerts numerous adverse effects on reproduction. The latest research has demonstrated that bisphenol A can induce the hypermethylation of CpG sites in the promoter region of *APOA1*, significantly elevating the transcription level of *APOA1* within the testes and enhancing testicular reverse cholesterol transport. Prolonged exposure to bisphenol A further increases HDL-C levels within the testes while significantly inhibiting TC and free cholesterol levels, leading to disruptions in sex hormone balance and the subsequent impairment of fish spermatogenesis [50].

## 3. APOC3: An Emerging Target in the Field of Lipid-Lowering Therapy

### 3.1. Function of APOC3

APOC3, which is primarily expressed in hepatocytes, is an apolipoprotein of 79 amino acid residues and is an important component of VLDL and HDL [51]. APOC3 has long been considered one of the most critical factors in TRL metabolism, which is closely related to the increase in the plasma TG level (Figure 1). As a co-factor of LPL, APOC3 is essential for regulating TRL lipolysis. The lipolysis efficiency of the LPL enzyme is related to the ratio of APOC2 and APOC3 on the particle surface [52]. APOC3 is a competitive inhibitor of the APOC2 activation of LPL, inhibiting LPL enzyme activity and preventing the degradation of TRLs [53]. APOC3 also blocks the binding of the LPL enzyme to glycosylphosphatidylinositol-anchored HDL binding protein 1 (GPIHBP1) on vascular endothelial cells. This prevents the anchoring of the LPL enzyme on the vascular endothelium, further amplifying APOC3’s inhibitory effect on triglyceride lipolysis [54]. In the liver, APOC3 facilitates VLDL assembly and secretion. The hepatic assembly of VLDL is achieved by recruiting large amounts of triglycerides as lipid droplets in the microsomal lumen to the VLDL precursors containing APOB-100. APOC3 promotes the binding of triglycerides to APOB-100 and the secretion of mature VLDL [55]. In addition, APOC3 displaces APOE from TRL or directly inhibits the binding of APOE to LDLR and LDLR-related protein 1 (LRP1) on the liver surface, thereby inhibiting the hepatic clearance of TRLs [56]. Notably, the CRISPR/Cas9 system was used to construct *APOC3*-KO rabbits, and knocking down *APOC3* in these rabbits led to a 50% decrease in the TG level with a significant increase in LPL activity. Moreover, APOC3 deficiency was found to be more beneficial in maintaining the TC, TG, and LDL-C levels, inhibiting inflammatory responses, and preventing atherosclerosis when fed a high-fat diet [57].

### 3.2. Epigenetic Regulation and Therapeutic Potential of APOC3

*APOC3* is the most important regulator in triglyceride metabolism, which further affects glucose and lipid metabolism, atherosclerotic plaque formation, inflammation, and endothelial endoplasmic reticulum stress response. *APOC3* is an important factor in the development of lipodystrophy, diabetic dyslipidemia, and coronary artery calcification [58,59,60,61]. Epidemiological studies and observational research have shown that increased levels of *APOC3* lead to increased TG levels and a high risk of ASCVD, while LOF mutations in *APOC3* are linked to decreased plasma TG levels and reduced cardiovascular disease risk among high-risk individuals [62,63]. For example, rs4225 in the 3’UTR region of the *APOC3* gene is the small ribonucleic acid binding site. The T allele of rs4225, but not the G allele, inhibited *APOC3* expression by promoting the binding of miR-4271 to the 3’UTR of *APOC3* mRNA. Individuals with the GG genotype exhibited elevated plasma APOC3 levels compared to individuals possessing the TT genotype. Patients with the G allele have a higher multivessel incidence and a greater incidence of left anterior descending artery disease, thus showing more severe ASCVD, which provides evidence that miR-4271 regulates *APOC3* expression levels and lipid metabolism [64] (Table 1).

Given that *APOC3* is an important target for reductions in TG and ASCVD risk, RNA interference therapies such as ASO and small interfering RNA (siRNA) techniques have been used to target *APOC3* mRNA [70]. ASO binds to the target mRNA and hydrolyzes the RNA strand by activating RNase. Different from ASO, double-stranded siRNA separates into two strands in the cytoplasm, with one strand hybridizing and degrading mRNA. N-acetyl galactosamine (GalNAc) is a carbohydrate with a high affinity to receptors on hepatocytes. Current studies often combine ASO or siRNA with GalNAc to improve the specific uptake of the liver so as to improve the efficacy under the condition of an equal dose and limit the off-target side effects [71]. ARO-APOC3, an siRNA drug targeting *APOC3*, has been proposed for the treatment of FCS. Phase 3 clinical trial results show that after 16 weeks of intervention, ARO-APOC3 treatment reduced LDL-C by 53% and TG by 55% without serious adverse events [72,73]. Volanesorsen, an anti-APOC3 ASO drug, has also entered the clinical trial stage. The results from phase 3 clinical trials demonstrate that Volanesorsen exhibits significant efficacy in reducing TG levels and decreasing the occurrence of acute pancreatitis events among patients with HTG. Consequently, it has received approval for treating FCS [74,75]. Another novel GalNAc-conjugated ASO drug, Olezarsen, has not only been proven to reduce TG levels in individuals with mild TG elevation but has also demonstrated good efficacy in patients with TG levels ranging from 200 to 500 mg/dL and a high risk of ASCVD. It has been shown to significantly reduce the risk of cardiovascular disease in patients with moderate to severe HTG with excellent safety and tolerability [71,76].

In extravascular lipid metabolism disorders, targeting *APOC3* therapy also has significant efficacy. An assessment of the hepatic fat fraction (HFF) in patients with severe HTG, familial partial lipodystrophy, and FCS after 12 months of Volanesorsen treatment showed that, compared with the controls, the HFF of the treatment group was significantly reduced by 3.02–8.34%. These results indicate that inhibiting hepatic *APOC3* synthesis has a good therapeutic effect on HFF [77].

Recently, numerous studies have reported epigenetic modifications of DNA and RNA targeting *APOC3*. DNA methylation marks of key genes are closely linked to inflammatory cell drive, apoptosis, thrombosis, and atherogenic signaling, as a recent study in a Chinese population with acute coronary syndrome has shown [78]. The identification of *APOC3* promoter methylation levels in patients with CAD showed that the CpG islands in the promoter region were hypermethylated [79]. In addition, METTL3 enhanced *APOC3* mRNA stability, increased the *APOC3* transcription level, and induced APOC3 protein expression in an m^6^A-dependent manner, while si-METTL3 reduced *APOC3* mRNA stability and then inhibited APOC3 protein expression [80]. Taken together, these studies provide evidence of the DNA methylation status and m^6^A modification of the *APOC3* gene (Figure 2).

## 4. APOA4: Biomarker for Disorders of Lipid Metabolism

### 4.1. Function of APOA4

APOA4 is a plasma lipoprotein that is primarily synthesized by the liver, as well as the small intestine, and subsequently secreted into the bloodstream (Figure 1). Plasma APOA4 is mainly distributed on the surface of chylomicrons and HDL, or it exists in a lipoprotein-free form. APOA4 is composed of 12 amphipathic helices that mediate lipid binding and interactions with aqueous humor. These helices are arranged to create a central hydrophobic pocket that can accommodate lipids, making them both adsorptive and exchangeable [81].

APOA4 is involved in lipid metabolism and glucose metabolism. As an exchangeable apolipoprotein, APOA4 participates in the secretion and clearance of chylomicrons and regulates the absorption of dietary fat [82]. Common APOA4 mutations include N147S, T347S, and Q360H [83]. A study by Hockey et al. showed that the 360H allele has greater lipid affinity compared with 360Q [84]. APOA4 also has an anti-atherosclerotic effect. In *APOE^−/−^* mice, a plasma lipid measurement and a quantitative analysis of aortic lesions indicated that the high expression of APOA4 in the intestine can reduce blood lipids and inhibit oxidative damage in local arterial tissues, thereby delaying the progression of atherosclerosis [85,86,87]. APOA4 also ameliorates hyperglycemia by increasing insulin secretion and glucose uptake activation in adipocytes. Further studies showed that APOA4 inhibited hepatic steatosis by down-regulating SREBF1-mediated lipogenesis and improved hepatic insulin sensitivity through IRS-PI3K-Akt signaling, thus ameliorating NAFLD [88]. In addition, APOA4 also plays an important anti-inflammatory role. Related studies have shown that APOA4 powerfully inhibits ROS activity. Moreover, the inhibition of APOA4 can suppress the activity of immune cells and reduce immune cell infiltration, further indicating that APOA4 has a bi-functional effect in regulating inflammatory injury and immune cell infiltration [89]. The overexpression of APOA4 attenuated liver injury by inhibiting the secretion of liver fibrosis mediators and inflammatory factors in CCL4-induced liver injury in mice, controlling the levels of antioxidant enzymes, and reducing the proportion of pro-inflammatory monocytes [90,91]. Thus, APOA4 may be a potential new therapeutic target for treating liver injury.

### 4.2. The Epigenetic Regulation and Therapeutic Potential of APOA4

Concerning the epigenetic regulation of *APOA4*, the lncRNA *APOA4-AS* has been identified as a co-regulator of *APOA4* expression. *APOA4* genes and *APOA4-AS* have similar expression patterns. The expressions of both *APOA4-AS* and *APOA4* were found to be abnormally elevated in the livers of ob/ob mice and in patients with fatty liver disease. The knockout of *APOA4-AS* reduced *APOA4* expression and resulted in lower plasma triglyceride and cholesterol levels in ob/ob mice. Mechanistically, *APOA4-AS* directly interacts with HUR to stabilize *APOA4* mRNA, whereas HUR loss significantly reduces *APOA4* transcripts [92]. Another nuclear lncRNA, Lnc19959.2, binds specifically to the *APOA4* transcriptional repressor Purb and promotes Purb ubiquitination and degradation, leading to increased *APOA4* expression [93].

*APOA4*, as a biomarker, plays an important role in disease prediction. Stephens–Johnson syndrome (SJS) and toxic epidermal necrolysis (TEN) are rare but severe adverse drug reactions. A plasma proteomic analysis showed that *APOA4* expressed significantly differentially between the treatment and control groups, and a multivariate regression analysis showed that the *APOA4* levels were highly associated with the prognostic parameters of SJS/TEN. Therefore, *APOA4* can be used as a prognostic marker for SJS/TEN [94]. Of note, *APOA4* also serves as a valuable predictor of the residual risk for coronary heart disease (CHD) and guides emerging preventive therapies against CHD. In the PROCARDIS CHD case–control study, the association between apolipoproteins and CHD risk was quantified using a single mass spectrometry assay. An elevated *APOA4* level was inversely associated with CHD risk, which was confirmed to be an associated factor for CHD independent of HDL. The detection and identification of *APOA4* will help to lower the residual risk in patients with CHD [95].

The pathogenesis of NAFLD is complex. Epigenetic modifications, especially DNA methylation changes, have been intensively studied in recent years [96,97]. *APOA4* exhibited DNA hypomethylation and had significantly higher expression in the liver of HFD-treated mice. This suggests that the assessment of the *APOA4* DNA methylation status and gene expression can be used to diagnose NAFLD and its severity [98] (Figure 2).

## 5. APOA5: A Regulator of Obesity and Metabolic Syndrome

### 5.1. Function of APOA5

APOA5 has been recognized as one of the most potent factors affecting plasma TGs [99]. APOA5, synthesized by the liver, is important for the maturation and secretion of VLDL and facilitates the formation of hepatic lipid droplets (LDs) [100,101]. Notably, APOA5 exerts an important function of lowering plasma triglycerides despite its low concentration (Figure 1).

Hepatic APOA5 can use fatty acids derived from adipocytes to promote the synthesis of LDs, thereby promoting lipid deposition in hepatocytes [102]. A series of clinical and in vivo experiments have proven that APOA5 can play a positive role in NALFD, and knocking down the intracellular expression of APOA5 leads to a significant reduction in intracellular TG. In the development of insulin resistance and obesity, APOA5 may act as a sensor for fatty acid accumulation in adipocytes, resulting in an increased number and increased size of LDs in the liver [103,104]. Thus, APOA5 may serve as an important regulator of TG storage in hepatocytes [105,106].

In plasma, APOA5 on the surface of TRLs enhances triglyceride hydrolysis and cholesterol remnant clearance. On one hand, APOA5 interacts with the ANGPTL3/8 complex and selectively blocks the inhibitory effect of the ANGPTL3/8 complex on LPL in a concentration-dependent manner [107]. On the other hand, APOA5 can also interact with glycosylphosphatidylinositol-anchored high-density lipoprotein binding protein 1 (GPIHBP1) to regulate LPL activity. GPIHBP1 is an endothelial membrane protein that facilitates the transfer of LPL from LPL-producing cells, including myocytes and adipocytes, to the vascular surface of the capillary endothelium, allowing LPL to function on the vascular surface. APOA5 interacts with GPIHBP1 to help TRLs adhere to the endothelial cell surface, and it promotes LPL-mediated TG hydrolysis [108]. Moreover, circulating APOA5 can also bind to the LDLR family and heparan sulfate proteoglycan (HSPG) family to effectively promote the removal of cholesterol remnants from circulation [109,110]. In conclusion, APOA5 maintains lipid homeostasis through multiple dimensions, and it especially plays a key role in TG homeostasis [100].

### 5.2. Epigenetic Regulation and Therapeutic Potential of APOA5

Many epigenetic factors, especially miRNA, regulate the expression of *APOA5*. It has been reported that the C allele of rs2266788 of the *APOA5* gene can generate a potential microRNA-binding site in the 3’UTR of *APOA5*, which is specifically recognized by liver-expressed miR-485-5p, leading to the down-regulation of the hepatic transcription of *APOA5* and the increase in the plasma TG level. Similarly, miR-3201 specifically bound to the T allele of rs2266788 and negatively regulated the transcriptional activity of *APOA5*. Reversely, the inhibition of miR-3201 expression significantly increased *APOA5* expression in HepG2 cells [68] (Table 1).

As a key molecule in triglyceride metabolism, the epigenetic heterogeneity of *APOA5* partially explains individual susceptibility to HTG. In patients with HTG, the *APOA5* promoter and exon 3 were hypermethylated. There was a significant positive correlation between exon 3 methylation and TG levels, and it is also positively associated with atherosclerotic dyslipidemia. Furthermore, the methylation rate of exon 3 exhibited a remarkable prevalence of 82% in patients with HTG harboring *APOA5* SNPs. Collectively, exon 3 CGI methylation in *APOA5* acts synergistically with genetic polymorphisms to increase the risk of HTG [111].

Recently, the prevalence of childhood obesity has been increasing, which has become a public health issue. Methylation is a key regulator of gene–environment interactions, and it is closely related to obesity. A genome-wide methylation array analysis of patients with obesity and controls revealed a significant inverse association between the level of methylation within the *APOA5* locus and obesity. It can be concluded that altered methylation at the CpG sites of specific genes, especially the altered methylation of genes regulating lipoprotein expression, may lead to childhood obesity. It also provides a new understanding of the etiology of obesity [112] (Figure 2).

## 6. Conclusions

Dyslipidemia, defined as high levels of lipids (TC, TG, or both) or low levels of HDL-C, directly increases the risk of many diseases including atherosclerosis, fatty liver disease, and acute pancreatitis. Among them, ASCVD deserves our attention. The effective treatment and management of this patient population has been a major challenge for the global medical community for many years due to the potential risk of high-risk cardiovascular events. Despite the great achievements of LDL-C-lowering drugs in reducing the risk of ASCVD, dyslipidemia characterized by elevated TRL levels still contributes to substantial residual cardiovascular risk and is increasing worldwide due to obesity and aging [17]. Of note, addressing the residual cardiovascular risk caused by high TRLs is a new direction for lipid management in the future.

Since the beginning of the 21st century, the development of gene therapy technologies, from traditional gene replacement to gene editing, has opened up endless possibilities for the treatment of dyslipidemia diseases. Besides gene editing systems such as CRISPR/Cas9, emerging epigenetic modulation techniques such as ASO and RNAi therapy are showing promise in regulating lipid homeostasis. The level of epigenetic modification of related genes can also play a role in the prediction and early warning of lipid metabolism disorders. Moreover, most phenotypically relevant sites were identified and translated into diagnostic assays for routine use in clinical utility along with advanced mapping methods for DNA and RNA modification. *APOA1/C3/A4/A5* plays an important role in lipid metabolism. Recent insights into *APOA1/C3/A4/A5* epigenetic mechanisms have increased our understanding of lipid metabolism. These findings provide new ideas for the prevention of lipid disorders and the reduction in residual cardiovascular risk, provide more effective and low-risk treatment methods, and fundamentally reduce the global economic burden associated with lipid metabolism disorders.

## Figures and Tables

**Figure 1 biomedicines-12-01224-f001:**
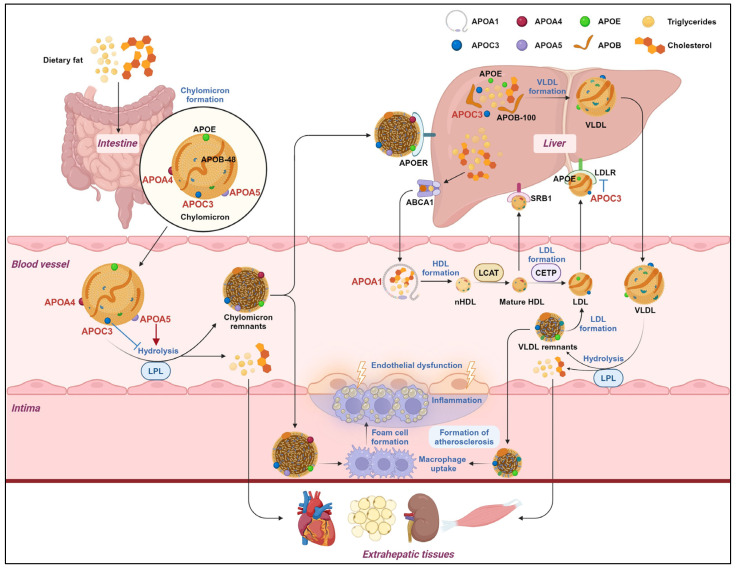
Dietary triglycerides and cholesterol are absorbed in the gastrointestinal tract and enter small intestinal epithelial cells. Cholesterol and triglycerides are assembled into CM through a synthetic pathway dependent on a microsomal triglyceride transporter and APOB-48 with the assistance of APOE, APOC3, APOA4, and APOA5. Endogenous triglyceride and cholesterol are synthesized in the liver and then closely combined with APOB-100, APOE, APOC3, and so on to assemble into VLDL. CM and VLDL are synthesized and transported into the blood and circulated, where they are hydrolyzed by LPL to generate fatty acids, VLDL remnants, and CM remnants. The fatty acids are internalized and used as an energy source in extrahepatic organs or for the storage of energy in adipocytes. Part of the CM remnants is ingested by the APOE receptor (APOER) and broken down in the liver, while part of the VLDL remnants is further converted to LDL. ABCA1, located on the surface of the peripheral cell membrane, transports excess free cholesterol and triglyceride into the circulating blood, which then combines with APOA1 to form nascent high-density lipoprotein (nHDL). Subsequently, nHDL is converted to spherical mature high-density lipoprotein (mature HDL) by esterification with LCAT. Mature HDL can be removed by SR-BI, or it can further transfer cholesterol to LDL by CETP. LDL produced by endogenous and exogenous pathways can be recognized and taken up by LDLR and then enter the liver for catabolism. APOC3 on the surface of LDL can displace APOE or directly interfere with the binding of APOE to LDLR, thereby inhibiting the hepatic clearance of LDL. The abnormal accumulation of CM and VLDL remnants in the subendothelium of the impaired arteries induces the production of various cytokines, inflammatory mediators, and biological enzymes, which promote the differentiation of macrophages. Differentiated macrophages phagocytize lipid particles and become foam cells, which die to form lipid pools and eventually develop into atherosclerotic plaques.

**Figure 2 biomedicines-12-01224-f002:**
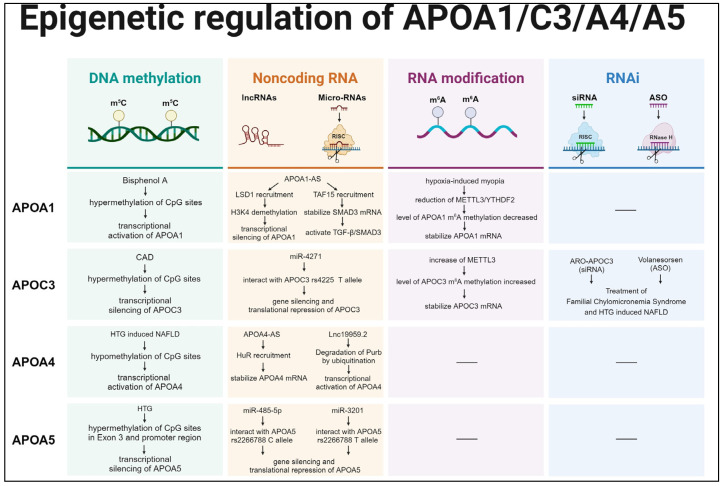
Epigenetic regulation patterns of *APOA1/C3/A4/A5* gene cluster.

**Table 1 biomedicines-12-01224-t001:** A summary of the effects of microRNAs on *APOA1/C3/A4/A5*.

Author	Gene	MicroRNA	Study Design	Major Outcomes
Fotini Kostopoulou et al. [65]	*APOA1*	MicroRNA-33a	The treatment of human normal chondrocytes with miR-33a	Reduced *APOA1* mRNA expression levels; induction of cholesterol metabolism disorders and osteoarthritic phenotype in normal chondrocytes
Li et al. [66]	*APOC3*	MicroRNA-424-5p	Aortic smooth muscle cells were treated with miR-424-5p mimic	Silence of *APOC3*; the proliferation, migration, and inflammation of aortic smooth muscle cells are inhibited, and apoptosis is promoted
Hu et al. [64]	*APOC3*	MicroRNA-4271	Investigating the effect of *APOC3* variants on microRNA binding	MicroRNA-4271 binds to *APOC3* and inhibits its transcription to reduce CHD risk
Cui et al. [67]	*APOA4*	MicroRNA-34a	Investigating gene expression levels in the livers of miR-34a^−/−^mice after perfluorooctanoic acid (PFOA) exposure	Under PFOA treatment, PPAR significantly up-regulates *APOA4* expression, while microRNA-34a only plays a moderate role
Cui et al. [68]	*APOA5*	MicroRNA-3201	Investigating the effect of *APOA5* variants on microRNA binding	The rs2266788 C allele interferes microRNA-3201 binding to *APOA5*, resulting in increased *APOA5* expression levels and risk of CAD
Cyrielle Caussy et al. [69]	*APOA5*	MicroRNA-485-5p	Investigating the effect of *APOA5* variants on microRNA binding	The rs2266788 C allele mediates microRNA-485-5p binding to *APOA5*, resulting in downregulation of *APOA5* and hypertriglyceridemic effect

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
