# Peer review of "APOA1/C3/A4/A5 Gene Cluster at 11q23.3 and Lipid Metabolism Disorders: From Epigenetic Mechanisms to Clinical Practices"

_biomedicines, 2024, doi:10.3390/biomedicines12061224_

Round 1

Reviewer 1 Report

Comments and Suggestions for Authors

This review investigates the biology of how the APOA1/C3/A4/A5 cluster affects circulating lipoproteins and the current progress in epigenetic regulation of the APOA1/C3/A4/A5 cluster and discusses emerging strategies for treating lipid metabolism disorders. The review is comprehensive and prospective. But I have some suggestions and questions:

1.      Introduce Figure 1 and Figure 2 in the main text.

2.      Will the epigenetic regulation of APOA1/C3/A4/A5 influence each other?     

In summary, I recommend this paper be minorly revised before acceptance.

Reviewer 2 Report

Comments and Suggestions for Authors

As a researcher who has conducted many GWAS studies related to metabolic diseases, this review was an opportunity to improve my understanding. Thank you for the opportunity to review these review papers.

Major comment

1. It would be helpful to understand the cluster area (11q23.3) accurately by briefly reviewing BUD13, ZPR1, and SIK1 as genes related to lipid metabolism around the area. 

2. It is necessary to revise the title to the review of the Chromosome 11q23.3 area.

3. APOA1/C3/A4/A5 functions and epigenetic regulations have been well reviewed. However, the reason why this cluster has become more interested in lipid metabolism is that it is the area where the most significant genetic indicators are found in most GWAS studies for lipid related phenotypes. Therefore, if author review this cluster, it seems that additional explanation is needed for the major results of GWAS, disease association, race difference, etc.

Minor comment

1. Line 39: chromosome 11q23 should be more accurately represented as 11q23.3.

Reviewer 3 Report

Comments and Suggestions for Authors

This manuscript summarises the role of APOA1/C3/A4/A5 in lipid (patho)physiology with emphasis on the epigenetic events that regulate this processes. The review is well written and the authors provide a comprehensive overview of the current literature with details appropriate to understand the implications of the dysregulation of APOA1/C3/A4/A5 gene expression which can potentially lead to the pathogenesis of lipid metabolism disorders.

It is strongly suggested that a section for the effects of microRNAs on this procedure should be added involving studies in humans, animals or cells. In addition, the incorporation of a summative table describing important in vivo and in vitro studies and their findings which contributed building the knowledge in this topic is also recommended in order to have a completed overview.
